# Tailoring Therapeutic Strategies in Non-Small-Cell Lung Cancer: The Role of Genetic Mutations and Programmed Death Ligand-1 Expression in Survival Outcomes

**DOI:** 10.3390/cancers15215248

**Published:** 2023-10-31

**Authors:** Nobuaki Kobayashi, Kenji Miura, Ayami Kaneko, Hiromi Matsumoto, Kohei Somekawa, Tomofumi Hirose, Yukihito Kajita, Anna Tanaka, Shuhei Teranishi, Yu Sairenji, Hidetoshi Kawashima, Kentaro Yumoto, Toshinori Tsukahara, Nobuhiko Fukuda, Ryuichi Nishihira, Makoto Kudo, Naoki Miyazawa, Takeshi Kaneko

**Affiliations:** 1Department of Pulmonology, Yokohama City University Graduate School of Medicine, Yokohama 236-0004, Japan; 2Department of Respiratory Medicine, Yokohama Sakae Kyosai Hospital, Yokohama 247-8581, Japan; 3Department of Pulmonology, Yokohama City University Medical Center, Yokohama 232-0024, Japan; 4Department of Respiratory Medicine, Kanto Rosai Hospital, Kawasaki 211-8510, Japan; 5Department of Respiratory Medicine, Yokohama Minami Kyosai Hospital, Yokohama 236-0037, Japan; 6Department of Respiratory Medicine, Chigasaki Municipal Hospital, Chigasaki 253-0042, Japan; 7Department of Respiratory Medicine, Fujisawa Municipal Hospital, Fujisawa 251-8550, Japan; 8Department of Respiratory Medicine, Yokohama Nanbu Hospital, Yokohama 234-0054, Japan

**Keywords:** non-small-cell lung cancer, molecular targeting agents, immune checkpoint inhibitors, overall survival, driver oncogene, PD-L1 expression, personalized medicine, real-world evidence

## Abstract

**Simple Summary:**

This study focuses on understanding the real-world impact of newly introduced genetic tests and immune-based approaches on the survival of patients with non-small-cell lung cancer (NSCLC). Conducted in multiple medical centers in Japan, this study involved 863 patients and evaluated various treatment methods, including targeted therapies and immune checkpoint inhibitors. Our results show that therapies tailored to specific genetic mutations led to significantly longer survival rates. Additionally, multivariate analysis identified the type of anticancer drug and the expression of programmed death-ligand 1 (PD-L1) at diagnosis as the impactful variables affecting 5-year OS. These findings underscore the importance of genetic and immune profiling in choosing the most effective treatment plans, thereby improving patient survival, and contributing to the advancement of personalized medicine in NSCLC.

**Abstract:**

Background: This study aims to assess the real-world impact of advancements in first-line systemic therapies for non-small-cell lung cancer (NSCLC), focusing on the role of driver gene mutations and programmed death-ligand 1 (PD-L1) expression levels. Methods: Conducted across eight medical facilities in Japan, this multicenter, retrospective observational research included 863 patients diagnosed with NSCLC and treated between January 2015 and December 2022. The patients were categorized based on the type of systemic therapy received: cytotoxic agents, molecular targeting agents, immune checkpoint inhibitors, and combination therapies. Comprehensive molecular and immunohistochemical analyses were conducted, and statistical evaluations were performed. Results: The median overall survival (OS) shows significant variations among treatment groups, with targeted therapies demonstrating the longest OS. This study also revealed that high PD-L1 expression was common in the group treated with immune checkpoint inhibitors. Multivariate analysis was used to identify the type of anticancer drug and the expression of PD-L1 at diagnosis as the impactful variables affecting 5-year OS. Conclusions: This study underscores the efficacy of targeted therapies and the critical role of comprehensive molecular diagnostics and PD-L1 expression in affecting OS in NSCLC patients, advocating for their integration into routine clinical practice.

## 1. Introduction

Lung cancer represents the most common cause of cancer-related mortality worldwide [1]. Among the various types of lung cancer, non-small cell lung cancer (NSCLC) is the most prevalent [2]. A significant challenge in the management of NSCLC is that the majority of cases are diagnosed at an advanced stage, necessitating the need for improvement in systemic therapy for recurrent and advanced stages of the disease [3].

The realm of advanced NSCLC has undergone remarkable transformations in this century, one of which is the discovery of driver gene mutations and the subsequent introduction of targeted molecular therapies [4]. Driver gene mutations are genetic alterations that are crucial for the initiation and maintenance of cancer. For instance, tyrosine kinase inhibitors targeting epidermal growth factor receptor (EGFR) mutations have dramatically improved the typical prognosis of advanced NSCLC from 12–14 months to 36–48 months [5,6,7,8]. Consequently, the identification of such mutations like anaplastic lymphoma kinase (ALK) [9], ROS proto-oncogene 1 (ROS1), V-raf murine sarcoma viral oncogene homolog B1 (BRAF) [10], MET proto-oncogene, receptor tyrosine kinase (MET) [11], and RET proto-oncogene receptor tyrosine kinase (RET) [12] has become increasingly critical in routine clinical practice.

Another pivotal development is the emergence of cancer immunotherapy. The discovery of immune checkpoint molecules such as programmed cell death protein 1 (PD-1) and cytotoxic T-lymphocyte-associated protein 4 (CTLA4) towards the end of the 20th century has led to an enhanced understanding of tumor immunosuppression [13,14]. Immune checkpoint inhibitors have augmented antitumor immunity, thereby incorporating immunotherapy into the standard treatment regimens for NSCLC [15].

While these revolutionary changes have reshaped clinical practice, their real-world impact remains uncertain. Clinical trial results often do not fully extend to the general patient population, which includes elderly individuals, those with poor performance status, and those with comorbidities. Therefore, this study aims to elucidate the impact of these new treatments on survival outcomes when employed as first-line therapy. Additionally, we intend to explore how the detection of these new driver gene mutations and the expression levels of PD-L1, which is a key player in tumor immunity, influence the prognosis of patients undergoing first-line systemic therapy.

## 2. Materials and Methods

### 2.1. Study Design

This is a multicenter, retrospective observational study conducted in collaboration with eight medical facilities located in Kanagawa Prefecture, Japan. It is important to note that all eight participating institutions are located within the same prefecture. The primary objective of this research is to assess the efficacy and safety of various systemic therapy as a first line in patients with NSCLC. Ethical approval for the study was granted by the Ethics Committee of Yokohama City University (Approval Number: B191200044). Informed consent was waived due to the retrospective nature of the study.

### 2.2. Study Population

Individuals eligible for this study were those who received a pathological diagnosis of non-small cell lung cancer (NSCLC) at any of the eight participating facilities. The diagnoses and treatments were conducted within the interval extending from 1 January 2015 to 31 December 2022. The inclusion criteria are further specified to encompass patients treated with any form of anti-cancer medication without restrictions based on performance status (PS). All enrolled patients in this study are Japanese. Exclusion criteria consisted of patients with lung cancer subtypes other than NSCLC, those undergoing chemoradiotherapy, and those receiving adjuvant chemotherapy following surgical intervention. However, patients who experienced disease recurrence following chemoradiotherapy or surgical resection were eligible for inclusion. This study observed patients from the initiation of their first-line treatment until the end of December 2022, serving as the final follow-up date.

The patients were stratified according to the type of systemic therapy received in the first line:

Cytotoxic agents (C): Medications such as cisplatin, carboplatin, paclitaxel, pemetrexed, gemcitabine, docetaxel, and S-1 were administered either intravenously or orally. Dose adjustments were implemented based on observed toxicities. No patients received both cytotoxic and molecular targeting agents concurrently.

Combination therapy (CI): regimens combining cytotoxic agents and immune checkpoint inhibitors (ICIs) were also administered.

Immune checkpoint inhibitors (I): Both monotherapies and combination therapies with different immune checkpoint inhibitors (ICIs) were included. In this study, monotherapy with immune checkpoint inhibitors was either pembrolizumab or atezolizumab. Combination therapy was conducted using nivolumab and ipilimumab.

Molecular targeting agents (M): EGFR tyrosine kinase inhibitors (TKIs), such as gefitinib, erlotinib, afatinib, and osimertinib, were used for patients with EGFR mutations. ALK inhibitors including crizotinib, alectinib, ceritinib, brigatinib, and lorlatinib were administered to patients harboring ALK rearrangements. Other targeted therapies, such as dabrafenib and trametinib for BRAF, crizotinib for ROS1, and tepotinib for MET mutations, were also employed.

### 2.3. Molecular and Immunohistochemical Analyses

Single-plex methods: A multiplex method is a technique that can simultaneously detect multiple targets in a single assay. These include polymerase chain reaction (PCR), fluorescence in situ hybridization (FISH), and immunohistochemistry (IHC). PCR was employed to amplify and detect specific sequences such as EGFR and ROS1, while FISH or IHC was used for detecting ALK rearrangements.

Multi-plex methods: A multiplex method is a technique that can simultaneously detect multiple targets in a single assay. Techniques like next-generation sequencing (NGS) based Oncomine Dx Target Test (ODx) (Thermo Fisher Scientific Inc., Waltham, MA, USA) or PCR-based AmoyDX (Amoy Diagnostics Co., Ltd., Xiamen, China) were employed. The test methodologies were validated and optimized at each participating facility, and the choice of method was determined by the facility or attending physician.

PD-L1 Expression: Immunohistochemical assays, including the 22C3 and 28–8 pharmDx assays (Dako North America, Inc., Carpinteria, CA, USA), as well as the SP142 and SP263 assays (Ventana Medical Systems, Inc., Tucson, AZ, USA), were used to quantify PD-L1 expression. The expression levels were reported as tumor proportion score (TPS), ranging from 0% to 100%. The categories were defined as follows: no (<1%), low (1–49%), and high (≥50%).

### 2.4. Statistical Analysis

The clinical data were extracted from electronic health records at each participating institution and sent to Yokohama City University for comprehensive analysis. Similarly, tests for driver oncogene mutations and PD-L1 expression were conducted at each institution using standardized and validated methods. These data were also collated at Yokohama City University for subsequent analysis. In the current study, overall survival (OS) was defined as the time interval from the initiation of treatment to the date of death from any cause. For patients whose vital status could not be confirmed, the survival time was censored at the last date of known contact or survival status update. Descriptive statistics, such as means, medians, and frequencies, were calculated. Inferential statistics employed included Kaplan–Meier survival analysis, log-rank tests, and multivariate Cox regression models. The data were analyzed using JMP version 17.0 (SAS Institute Inc., Cary, NC, USA). The level of statistical significance was set at *p* < 0.05. The survival curves were generated using Python (version 3.9.17), with Numpy (version 1.25.2) and Lifelines (version 0.27.7) for the statistical analysis.

## 3. Results

### 3.1. Impact of Molecular Targeted Agents and Immune Checkpoint Inhinitors

#### 3.1.1. Patient Characteristics

A total of 863 patients were included in this retrospective observational study across multiple institutions, as delineated in Figure 1. In Table 1, the patient characteristics reveal several features that distinguish the groups treated with different systemic anticancer treatments regimens (C: chemotherapy with cytotoxic agent, CI: combination chemotherapy with cytotoxic agents and immune checkpoint inhibitor, I: immune checkpoint inhibitor, M: molecular targeting agent). The M group has a notably higher number of females (215) compared to other groups. In contrast, the C and CI groups have a more balanced gender distribution. The median age of the participants varied across treatment groups, ranging from 69 to 75 years. The majority of the patients were in advanced stages of the disease.

The histological types were predominantly adenocarcinoma, followed by squamous cell carcinoma, consistent with the common histological subtypes usually observed in this cancer type. The M group has a significantly higher number of adenocarcinomas (Ad) compared to other pathologies. Sq (squamous cell carcinoma) is more prevalent in the C and CI groups, while it is almost absent in the M group. The metastatic profiles reveal that bone metastasis was the most common, followed by pleural and brain metastasis.

The M group has a strikingly high number of EGFR mutations, while ALK mutations are also notably higher in this group. The CI group has a higher prevalence of KRAS mutations. Regarding PD-L1 expression, high PD-L1 expression is notably common in the I group, which is likely due to the nature of immune checkpoint inhibitors targeting PD-L1. Low expression is more evenly distributed.

#### 3.1.2. Overall Survival among Patients Treated with Different Systemic Anticancer Treatments

The Kaplan–Meier survival curves in Figure 2 provide a comprehensive assessment of overall survival (OS) in patients with NSCLC who were treated with different systemic anticancer treatments regimens. The median OS for patients treated with immune checkpoint inhibitors (I) was 21.75 months with a 95% confidence interval (CI) of 11.07 to 36.57 months. In contrast, those treated with chemotherapy (C) had a median OS of 16.29 months (95% CI: 13.86–20.50). The combination therapy group (CI) demonstrated a median OS of 32.75 months, with the upper limit of the 95% CI extending to infinity. The molecular targeting agents group (M) showed the longest median OS, recorded at 49.93 months (95% CI: 36.75–58.07).

Statistical evaluations were performed using the log-rank test. The results show that the *p*-values for comparisons between I/M, C/CI, and C/M were less than 0.0004, indicating statistically significant differences in survival outcomes. On the other hand, the *p*-values for I/C and I/CI were 0.404 and 0.067, respectively. The hazard ratios (HRs) were calculated to further substantiate these findings. The HR for C compared to I was 1.1508 (95% CI: 0.8324–1.5910), for CI compared to I was 0.7164 (95% CI: 0.5000–1.0263), and for M compared to I was 0.4754 (95% CI: 0.3420–0.6610).

The data notably highlight the importance of identifying driver oncogene mutations and administering appropriate molecular targeting agents for improved patient outcomes. The strikingly long median OS in the M group underscores the efficacy of personalized, targeted therapies. Additionally, the use of immune checkpoint inhibitors demonstrates significant promise, particularly for a subset of patients, leading to improved long-term prognosis.

### 3.2. Impact of Driver Oncogene Mutation and Molecular Targeted Agents

#### 3.2.1. Prevalence of Driver Oncogene Mutations in the Cohort

Figure 3 provides the distribution of various driver oncogene mutations among the 863 patients in the study. The EGFR gene mutation was the most common, found in 310 patients (35.9%). This prevalence underscores its key role in NSCLC and the need for EGFR-targeted therapies. In contrast, no identifiable driver oncogenes were detected in 396 patients (45.9%). ALK mutation was next in frequency, identified in 30 patients (3.5%). Less prevalent mutations such as KRAS were found in 16 patients (1.9%), while MET and ROS1 mutations were observed in 11 patients each (1.3%). The least common of the known driver mutations was BRAF, detected in three patients (0.3%) of the study cohort. Notably, 86 patients, or about 10% of the study population, were not screened for driver oncogenes and were marked as “NE”.

#### 3.2.2. Comparative Analysis of Diagnostic Methodologies for Detecting Driver Oncogene Mutations in NSCLC

The heterogeneous nature of NSCLC necessitates a comprehensive approach to molecular diagnostics. In our cohort, we evaluated the efficacy of four diagnostic methods in detecting key driver oncogenes. Table 2 provides these methods ranked by the number of identified mutations for each oncogene. With 498 total tests conducted, the single methods, including PCR for EGFR and immune histochemistry for ALK, demonstrated considerable proficiency in detecting EGFR mutations, with a positive rate of approximately 43.9%. ALK and ROS1 mutations were also reasonably well-detected with positive rates of 4.21% and 1.61%, respectively.

A total of 192 assays were conducted using the Oncomine Dx (ODx) methodology. While its positive rate for detecting KRAS mutations was around 3.65%, it was used to detect EGFR mutations in 56 instances (29.1%). For ALK, six mutations were identified, translating to a positive rate of about 3.13%. Notably, these positive rates for EGFR and ALK are lower compared to those achieved with the single method, warranting careful consideration in clinical scenarios where these genes are of primary interest. Liquid biopsy was only performed to detect EGFR in this study. It detected EGFR mutations in six instances.

#### 3.2.3. Overall Survival following Molecular Targeted Therapy Based on Driver Oncogene Mutations

The impact of molecular targeted therapies on OS was evaluated in distinct patient subsets, each characterized by specific driver oncogenes. Figure 4A elucidates these findings.

Among patients with EGFR mutation, they had a median OS of 46.1 months, with a 95% confidence interval ranging from 33.7 to 57.8 months. Patients with ALK mutations displayed an exceptional median overall survival that was not calculable due to the majority of patients remaining alive at the last follow-up. The median OS values among patients with BRAF/MET/ROS1 were 21.6/27.1/37.5, respectively. Among the detectable KRAS mutations were various types, including G12C. However, as sotorasib, a molecular targeting agent for KRAS G12C, was not approved during the period of this research, a detailed breakdown of KRAS mutations is not provided.

While each subgroup demonstrated varying degrees of overall survival, no statistically significant differences were observed among them. This lack of statistical differentiation may partially be due to the limited number of patients in some subgroups, which could influence the power of the analysis. Nevertheless, the strong performance of targeted therapies, notably in the ALK and EGFR-positive groups, accentuates the clinical importance of molecular diagnostic profiling for effective patient management.

#### 3.2.4. Overall Survival among Patients without Driver Oncogene Mutations or Unevaluated for Mutations after Initiation of Cytotoxic Agents with or without Immune Checkpoint Inhibitors

Figure 4B indicates the OS outcomes in two distinct groups: those without any detected driver oncogene mutations (“None”) and those who were not evaluated for these mutations (“NE”). Patients in the NE group, who were not evaluated for driver oncogene mutations, had a median OS of 15.5 months. The 95% confidence interval for this estimate ranged from 11.3 to 21.1 months. Contrastingly, patients who were evaluated and found to have no driver oncogene mutations had a notably superior median OS of 23 months. The 95% confidence interval spanned from 17.1 to 30.3 months.

The significantly better OS in the None group, as compared to those not evaluated, underscores the importance of molecular diagnostic profiling. This difference in OS between the two groups illustrates the potential benefits of undergoing evaluation for driver oncogene mutations, even if none are detected. The findings emphasize the clinical utility of comprehensive genetic screening for tailoring therapeutic strategies, particularly when cytotoxic agents and immune checkpoint inhibitors are considered for treatment.

### 3.3. Impact of Expession of PD-L1 and Immune Checkpoint Inhibitors

#### 3.3.1. Overall Survival Stratified by PD-L1 Expression Levels in NSCLC Patients

Our study further scrutinized the role of PD-L1 expression levels in influencing the OS outcomes among NSCLC patients. Figure 5A delineates the Kaplan–Meier curves for OS across four distinct PD-L1 expression categories: high, low, not evaluated (NE), and no expression (no). Statistical analysis using the log-rank test yielded a significant *p*-value of 0.0167. This indicates the importance of PD-L1 expression levels in determining survival outcomes in NSCLC.

As expected, Figure 5B reveals that patients with high PD-L1 expression demonstrated better OS when treated with immune checkpoint inhibitors, as opposed to their outcomes in Figure 5C,D. Surprisingly, the OS among patients with no PD-L1 expression was similar to that of patients with high PD-L1 expression when treated with immune checkpoint inhibitors. This nuanced observation could be a contributing factor to why patients with low or no PD-L1 expression exhibited better OS compared to those with high PD-L1 expression in Figure 5A.

#### 3.3.2. Multivariate Analysis of Factors Affecting Overall Survival in NSCLC Patients

A comprehensive multivariate analysis was undertaken to identify the key determinants affecting OS. Variables considered in the analysis included the type of anticancer drug administered, PS at diagnosis, presence of bone, liver, brain metastasis, PD-L1 expression levels, and age (Table 3).

Among the variables analyzed, the type of anticancer drug used, categorized as C, CI, I, or M, emerged as a highly significant factor affecting 5-year OS. The L-R ChiSquare value for this variable was 68.51, with a *p*-value of less than 0.0001, and a logworth of 14.050. Similarly, the patient’s performance status (PS) at the time of diagnosis was another critical variable. PD-L1 expression levels were also found to be of statistical importance but to a lesser extent than the aforementioned variables. In contrast, liver metastasis, age, and brain metastasis were not statistically significant in this study.

The type of anticancer drug and PS at diagnosis were identified as the most impactful variables affecting the 5-year OS in lung cancer patients. While other factors like bone metastasis and PD-L1 expression also play a role, they are of secondary importance in the context of this study.

Firstly, the clinical stage of the tumors was reported according to the eighth edition of the Union for International Cancer Control (UICC) / American Joint Committee on Cancer (AJCC) staging system. For patients initially staged using the seventh edition, reclassification was performed in accordance with the updated eighth edition criteria. Cases at any Stage IV were grouped collectively and denoted simply as Stage IV. C: chemotherapy with cytotoxic agent, CI: combination systemic anticancer treatments with cytotoxic agents and immune checkpoint inhibitor, I: immune checkpoint inhibitor, M: molecular targeting agent, Ad: adenocarcinoma, Sq: squamous cell carcinoma, NOS: not otherwise specified, Post op: postoperative recurrence, EGFR: epidermal growth factor receptor, ALK: anaplastic lymphoma kinase, KRAS: Kirsten rat sarcoma viral oncogene homolog, MET: mesenchymal–epithelial transition factor, ROS1: ROS proto-oncogene 1, BRAF: B-Raf proto-oncogene, NE: not evaluated.

## 4. Discussion

This study provides a comprehensive evaluation of the factors influencing OS among patients diagnosed with NSCLC, incorporating a wide spectrum of variables such as types of systemic therapy, driver oncogene mutations, and PD-L1 expression levels in the real-world settings. The data gathered across multiple institutions lend a robust foundation to our findings, which have significant implications for both clinical practice and future research. The current data offer an invaluable contribution to emphasize the revolutionary advancements made in this century concerning genetic screening, molecularly targeted therapies, and immuno-modulation through PD-L1 expression. This is in line with the findings of the study by Danesi V. et al., which also emphasizes the importance of real-world evidence in understanding the impact of first-line immunotherapy in treating metastatic NSCLC [16]. These innovations have not only refined our understanding of NSCLC, but also have opened new avenues for personalized and effective treatment strategies.

Our results underscore the fundamental role of molecular targeted therapies in improving the 5-year OS in patients with NSCLC, corroborating the findings of Lee, D.H. in the PIvOTAL observational study, which also highlights the significance of molecular testing for optimizing treatment strategies [17]. Molecular targeting agents, specifically those targeting EGFR [6,18], ALK [19,20,21], ROS1 [22], and BRAF [23], have shown superior outcomes in clinical trials involving patients with favorable conditions, surpassing the results achieved by standard therapies. Our findings corroborate other similar reports [24,25,26,27], emphasizing the necessity of comprehensive genetic testing to identify appropriate candidates for molecular targeted therapies in a more general patient population. Despite the initial efficacy, resistance mechanisms often emerge, particularly in the context of therapies targeting EGFR [28,29] and ALK [30,31]. The development of next-generation inhibitors and combination therapeutic strategies may provide avenues for surmounting acquired resistance, thereby further extending the survival benefits conferred by molecular targeted treatments [32,33].

We compared four methods for detecting driver mutations in NSCLC: single, ODx, liquid biopsy, and others (Figure 3 and Table 2). The single method, using PCR and immunohistochemistry, was the best for EGFR (43.9%) and ALK (4.21%). ODx was the best for KRAS (3.65%). NGS was the best for ROS1 (2.9%). Liquid biopsy only detected EGFR (6.9%). The single method has several advantages over other methods, such as cost-effectiveness and providing rapid results within a few hours, which is crucial for timely initiation of targeted therapy [34]. However, the limitations of the single method are limitation for number of mutations in one assay. In this study, HER2, RET, or NTRK were not screened using the single method. Thus, multiplex methods to screen oncogene mutations, including ODx, are recommended by several guidelines [35,36]. Multiplex methods may have lower sensitivity or specificity than single-gene methods for certain mutations [37]. Multiplex methods may generate false-positive or false-negative results due to technical errors or biological heterogeneity. Therefore, multiplex methods should be carefully validated and quality-controlled before clinical application [38].

Our study highlights the multifaceted impact of PD-L1 expression in lung cancer. Essentially, high PD-L1 levels can benefit patients on immune therapies but may complicate the effectiveness of other treatments (Figure 5 and Table 3). The role of PD-L1 expression as a predictive factor for the efficacy of cytotoxic anti-cancer agents is less clear-cut. Despite this uncertainty, it is generally understood that the presence of PD-L1 expression, by virtue of its immune-suppressive effects, could be a poor prognostic factor for treatments other than immune checkpoint inhibitors. This understanding is further nuanced by the relationship between PD-L1 and specific genetic mutations, such as EGFR mutations. High PD-L1 expression has been reported as an adverse predictor for treatment response in EGFR-positive lung cancer treated with EGFR-TKIs [39,40,41].

While this study provides substantive insights into the modern management of NSCLC, several limitations should be acknowledged to contextualize the findings. First, the retrospective nature of this study limits causal interpretations and may introduce selection bias and unmeasured confounding variables. Second, the patient cohort was heterogeneous, consisting of diverse disease stages and treatment histories. Third, the sample size was particularly small for rare mutation subgroups like ALK, MET, and ROS1, which affects the statistical power of the study. Fourth, multiple diagnostic methods for detecting driver oncogenes added a layer of complexity to the results. Importantly, not all patients in the study were uniformly screened for NTRK, HER2, and RET mutations. This absence could limit the comprehensiveness of our genetic screening and may have implications for the generalizability of our findings. Fifth, the study primarily utilized PFS and OS as the metrics for assessing therapeutic efficacy. Although these are robust endpoints, other potentially useful metrics such as objective response rate and disease control rate were not included. Moreover, this study did not assess therapy-related quality of life, another critical indicator for evaluating the effectiveness of systemic anticancer treatments regimens. These limitations should guide future research, which could benefit from prospective, randomized designs with larger and more diversified patient populations.

In summary, a comprehensive approach to lung cancer treatment should consider both the genetic and immunological landscapes. PD-L1 expression could potentially serve as a biomarker for assessing treatment efficacy across a range of therapies, albeit with varying implications depending on the specific genetic mutations present in the tumor. Our findings underscore the need for ongoing research into the interactions between PD-L1 expression, genetic mutations, and different treatment regimens. Such efforts will be crucial in paving the way for more personalized and effective therapeutic strategies, leveraging the insights from both molecular-targeted agents and immune modulators to improve outcomes for lung cancer patients.

## 5. Conclusions

In conclusion, this study emphasizes the critical role of genetic screening, targeted molecular therapies, and immune checkpoint inhibitors in the management of non-small cell lung cancer (NSCLC). Our data highlight the significance of comprehensive molecular diagnostics and PD-L1 expression in affecting overall survival, thereby advocating for their integration into routine clinical practice. These findings contribute to the burgeoning field of personalized medicine in NSCLC, underscoring the need for multidisciplinary approaches for optimized therapeutic outcomes.

## Figures and Tables

**Figure 1 cancers-15-05248-f001:**
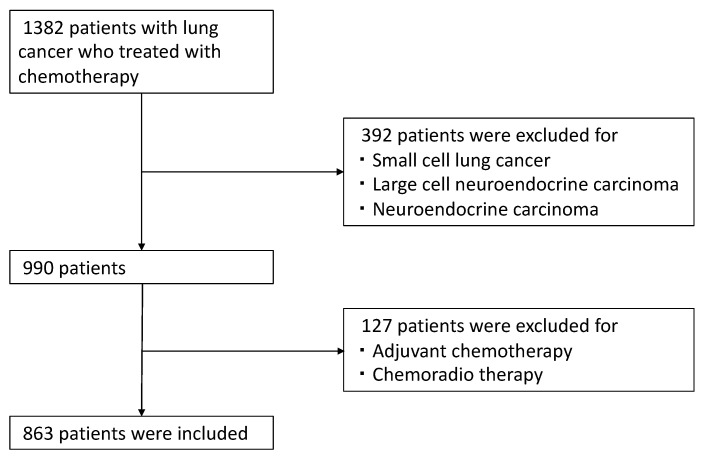
Screening, Exclusion, and Inclusion in the Study.

**Figure 2 cancers-15-05248-f002:**
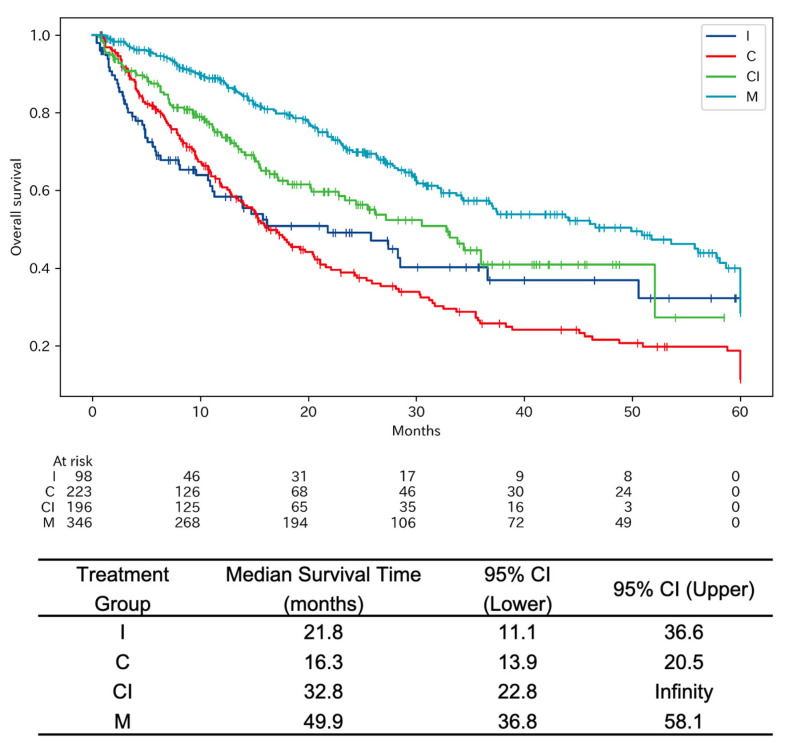
Kaplan–Meier survival curves for patients with NSCLC undergoing different systemic anticancer treatments. Figure 2 indicates the Kaplan–Meier survival curves for NSCLC patients who underwent various systemic anticancer treatment regimens. The treatment groups are categorized into four subgroups: I for immune checkpoint inhibitor, C for chemotherapy with cytotoxic agents, CI for combination chemotherapy with cytotoxic agents and immune checkpoint inhibitor, and M for molecular targeting agents. The median OS for group I was 21.75. For group C, the median OS was 16.29 months. Group CI had a median OS of 32.75 months, and group M had a median OS of 49.93 months. The statistical significance was assessed using the log-rank test. The *p*-values for comparisons between I/C, I/CI, I/M, C/CI, C/M, and CI/M were 0.404, 0.067, less than 0.00001, 0.001, less than 0.00001, and 0.0004, respectively.

**Figure 3 cancers-15-05248-f003:**
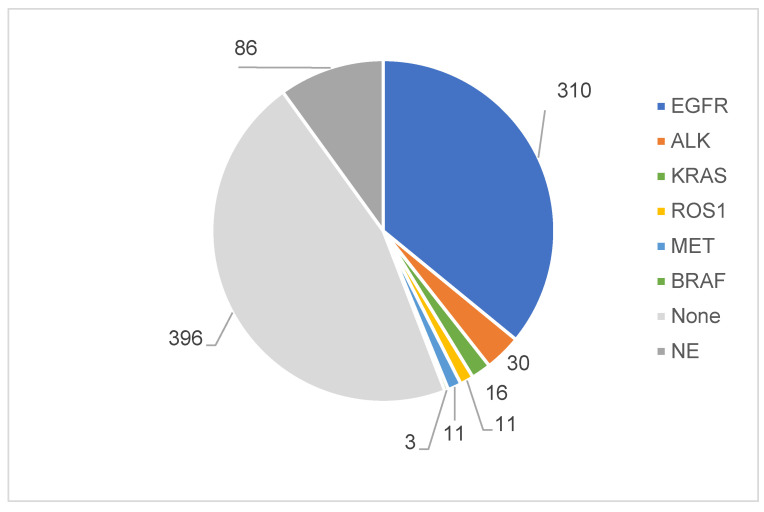
Distribution of driver oncogene mutations in a cohort of 863 NSCLC patients. The frequency and percentage of various driver oncogene mutations. The EGFR mutation was identified in 310 patients (35.9%). 396 patients (45.9%) had no detectable driver oncogenes (“None”). ALK mutations were found in 30 patients (3.5%), KRAS in 16 (1.9%), MET and ROS1 each in 11 (1.3%), and BRAF in 3 (0.3%).

**Figure 4 cancers-15-05248-f004:**
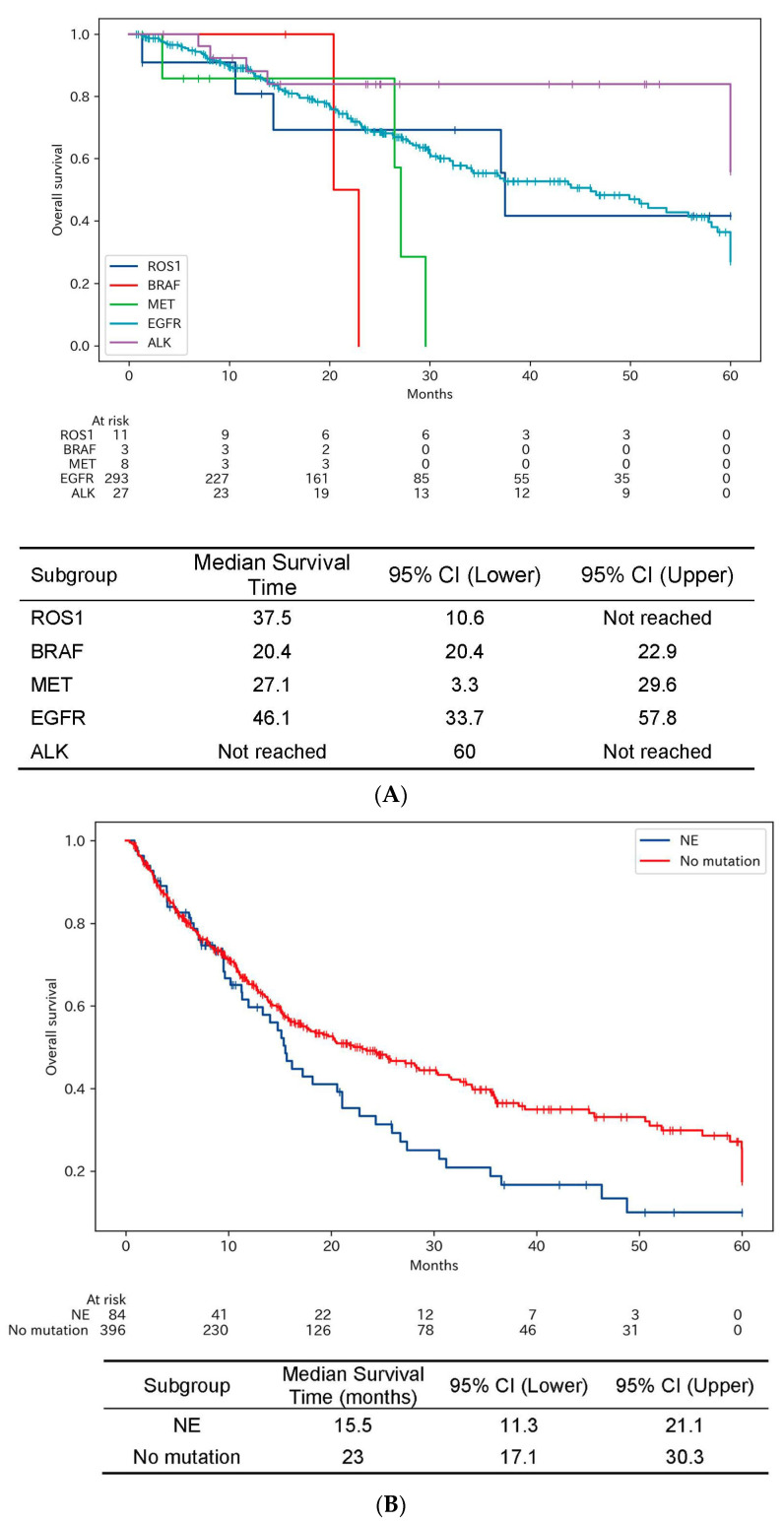
(**A**) Kaplan–Meier curves for overall survival after initiation of molecular targeting agents among patients with indicated driver oncogene mutations. The Kaplan–Meier survival curves showing OS outcomes for patients treated with molecular targeting agents, categorized based on the presence of specific driver oncogene mutations. The subgroups include ALK, BRAF, EGFR, MET, and ROS1. Notably, no statistical difference was observed in OS among the various subgroups, despite the limited number of patients in some categories. (**B**) Kaplan–Meier curves for overall survival after initiation of anti-cancer systemic therapy among patients without driver oncogene mutations or not evaluated for mutations. The Kaplan–Meier survival curves for two distinct patient groups: those who were not evaluated for driver oncogene mutations (“NE”) and those without any detected mutations (“No mutation”). Median survival times and 95% confidence intervals are given for each group. The figure highlights the significantly superior OS outcomes for patients who were evaluated and found to have no driver oncogene mutations, compared to those not evaluated (*p* = 0.021).

**Figure 5 cancers-15-05248-f005:**
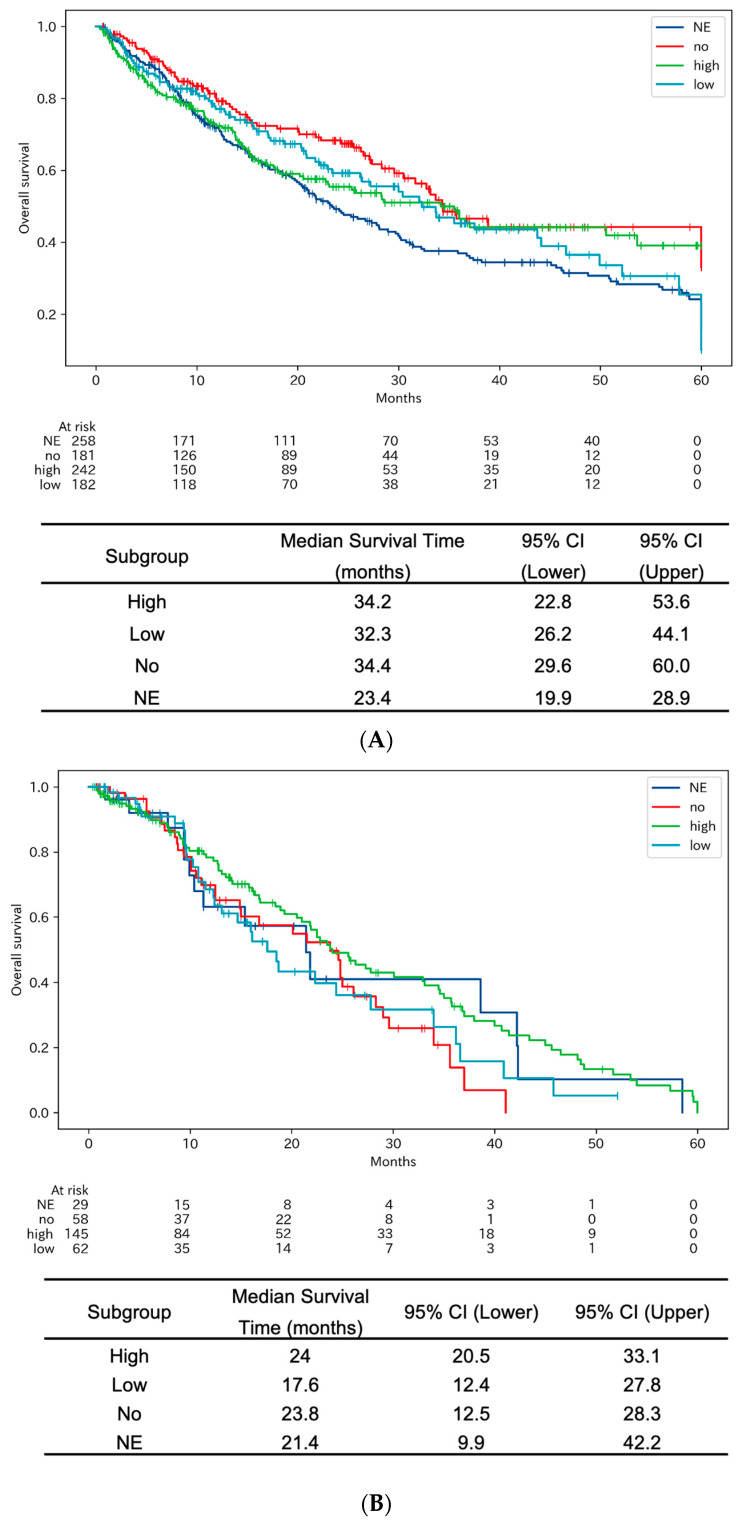
Kaplan–Meier curves for overall survival after initiation of systemic anticancer treatments among patients with indicated PD-L1 expression levels. (**A**) Among patients treated with any form of treatment. (**B**) Among patients treated with combination chemotherapy and immune checkpoint inhibitors or immune checkpoint inhibitors alone. (**C**) Among patients treated with molecular targeting agents. (**D**) Among patients treated with chemotherapy using cytotoxic agents.

**Table 1 cancers-15-05248-t001:** Patient characteristics.

	C	CI	I	M
N	223	196	98	346
Gender				
Female	54	37	28	215
Male	169	159	70	131
Age				
Smoking habit				
Never	34	18	15	188
Ex-smoker	115	103	53	112
Current	74	73	29	44
Performance status				
0	77	74	20	131
1	116	102	54	153
2	26	18	23	43
3	4	1	1	16
4	0	1	0	2
Median (min–max)	72 (23–86)	69 (39–85)	75 (36–88)	72 (27–93)
Pathology				
Ad	128	129	51	327
Sq	67	47	33	5
NOS	25	13	12	12
Ad + Sq	2	1	0	1
Large	0	0	0	1
Other	1	6	2	0
Clinical stage				
IIB, IIIA	13	3	3	8
IIIB	30	9	12	17
IV	141	143	68	254
Post op	37	40	15	67
Metastasis location				
Bone	55	56	22	120
Pleura	40	54	25	82
Brain	27	35	18	82
Liver	15	18	9	35
Driver oncogene mutation				
EGFR	11	6	0	293
ALK	2	1	0	27
KRAS	1	13	2	0
MET	0	3	0	8
ROS1	0	0	0	11
BRAF	0	0	0	3
None	161	148	86	1
NE	48	25	10	3
PD-L1 expression				
High	35	62	83	62
Low	38	52	10	82
No	39	55	3	84
NE	111	27	2	118

**Table 2 cancers-15-05248-t002:** Results of Screening for Driver Oncogene Mutations by Diagnostic Method.

Diagnostic Method	EGFR	ALK	KRAS	ROS1	MET	BRAF	None	Total
Single (S)	219 (44.0%)	21 (4.2%)	3 (0.6%)	8 (1.6%)	4 (0.8%)	1 (0.2%)	242 (48.6%)	498
Oncomine Dx (ODx)	56 (29.3%)	6 (3.1%)	7 (3.7%)	2 (1.0%)	3 (1.6%)	1 (0.5%)	116 (60.7%)	191
Liquid (L)	6 (75.0%)	0 (0.0%)	0 (0.0%)	0 (0.0%)	0 (0.0%)	0 (0.0%)	2 (25.0%)	8
Others (OTH)	29 (36.2%)	3 (3.8%)	6 (7.5%)	1 (1.2%)	4 (5.0%)	1 (1.2%)	36 (45.0%)	80

**Table 3 cancers-15-05248-t003:** Multivariate Analysis of Factors Affecting Overall Survival in NSCLC Patients.

Variable/Factor	Degrees of Freedom	L-R ChiSquare	*p*-Value	Logworth
Type of anticancer drug (C, CI, I, M)	3	68.51	<0.0001	14.05
PS at diagnosis	1	44.72	<0.0001	10.643
Bone metastasis	1	7.96	0.0048	2.321
PD-L1 expression (no, low, high)	3	10.02	0.0184	1.734
Liver metastasis	1	3.03	0.082	1.086
Age	1	1.45	0.2279	0.642
Brain metastasis	1	1.01	0.315	0.502

PS: performance status.

## Data Availability

The data that support the findings of this study are not publicly available due to privacy and ethical considerations. Data are available from the authors upon reasonable request and with permission of the Ethical Committee of Yokohama City University.

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
