# Peer review of "Tailoring Therapeutic Strategies in Non-Small-Cell Lung Cancer: The Role of Genetic Mutations and Programmed Death Ligand-1 Expression in Survival Outcomes"

_cancers, 2023, doi:10.3390/cancers15215248_

Round 1

Reviewer 1 Report

Comments and Suggestions for Authors

This is an interesting article, alktohough no significant information is added to literature data. 

1. In study polulation section, the sentences lines 98-101 must be re-written: it is reported that patients received just chemotherapy. between 2015 and 2022. Inclusion criteria need to better detailed.

2. Separate paragraphs need to be specified: line 220 and 236.

3. Line 225: KRAS patients are not described, they are reported in table 1 and figure 3. Were they G12C positive? 

4. In the discussion, the limitations of the study need to be better described. For example, the lack of some tests (NRTRK, HER2 , RET) need to be highlighted.

Author Response

Response to Reviewer 1 Comments

1. Summary        
Thank you very much for taking the time to review this manuscript. Please find the detailed responses below and the corresponding revisions in track changes in the re-submitted files.

2. Point-by-point response to Comments and Suggestions for Authors

Comments 1: In study polulation section, the sentences lines 98-101 must be re-written: it is reported that patients received just chemotherapy. between 2015 and 2022. Inclusion criteria need to better detailed. 

Response 1: Thank you for pointing this out. I/We agree with this comment. To address your feedback, we have modified the "Study Population" section. The revised section now clarifies that eligible individuals were those pathologically diagnosed with non-small cell lung cancer (NSCLC) and treated within the interval from January 1, 2015, to December 31, 2022. Furthermore, the inclusion criteria have been refined to specify that patients who were administered any anti-cancer medication were eligible, thereby expanding the scope beyond chemotherapy alone. We trust that these modifications sufficiently address your concerns. 

In line 98 - 102 “Individuals eligible for this study were those who received a pathological diagnosis of non-small cell lung cancer (NSCLC) at any of the eight participating facilities. The diagnoses and treatments were conducted within the interval extending from January 1, 2015, to December 31, 2022. The inclusion criteria are further specified to encompass patients treated with any form of anti-cancer medication. without restrictions based on performance status (PS).”

Comments 2: Separate paragraphs need to be specified: line 220 and 236.

Response 2: Thank you for your comments. In compliance with your recommendation, we have separated the content at lines 220 and 236 into distinct paragraphs. This modification aims to enhance the readability of the manuscript and to delineate the specific points more clearly. 

Comments 3: Line 225: KRAS patients are not described, they are reported in table 1 and figure 3. Were they G12C positive?

Response 3: Thank you for drawing attention to the lack of specificity regarding KRAS mutations. Your observation is astute. In our study, various KRAS mutations, including G12C, were indeed detected. However, we opted not to provide a detailed breakdown of KRAS mutations because sotorasib, a molecular targeting agent for KRAS G12C, was not approved during the period of this study. Given this, we felt that such specificity would not add substantial value to the analysis presented. To clarify this point, we have included an explanatory note in the manuscript.

In line 239 – 242; “Among the detectable KRAS mutations were various types, including G12C. However, as sotorasib, a molecular targeting agent for KRAS G12C, was not approved during the period of this research, a detailed breakdown of KRAS mutations is not provided.”

Comments 4: In the discussion, the limitations of the study need to be better described. For example, the lack of some tests (NRTRK, HER2 , RET) need to be highlighted.

Response 4: Thank you for your insightful comment regarding the limitations of our study. We agree that a more detailed discussion about the limitations, especially the absence of tests for certain mutations like NTRK, HER2, and RET, would enhance the manuscript. We have revised the Discussion section to include these points, thereby providing a more nuanced context for interpreting our findings.

In line 428 – 437; “First, the retrospective nature of the study limits causal interpretations and may introduce selection bias and unmeasured confounding variables. Second, the patient cohort was heterogeneous, consisting of diverse disease stages and treatment histories. Third, the sample size was particularly small for rare mutation subgroups like ALK, MET, and ROS1, which affects the statistical power of the study. Fourth, multiple diagnostic methods for detecting driver oncogenes added a layer of complexity to the results. Importantly, not all patients in the study were uniformly screened for NTRK, HER2, and RET mutations. This absence could limit the comprehensiveness of our genetic screening and may have implications for the generalizability of our findings.”

Reviewer 2 Report

Comments and Suggestions for Authors

It is not clear that, in which institute the study was conducted. Patient sample were collected at eight places but at which institute the experiments was conducted is not mentioned in the manuscript.

Author should add a demographic data in the form of table or chart, to assess the type of mutation associated with the area where they enrolled.

In result, author should add legends with figure 4A to 4D.

Author may elaborate discussion with observed result and existing or used other techniques used to evaluate the efficacy of therapy.

Author Response

Thank you very much for taking the time to review this manuscript. Please find the detailed responses below and the corresponding revisions in track changes in the re-submitted files.

Point-by-point response to Comments and Suggestions for Authors

Comments 1: It is not clear that, in which institute the study was conducted. Patient sample were collected at eight places but at which institute the experiments was conducted is not mentioned in the manuscript.

Response 1: Thank you for your comment. We realize the necessity of clearly indicating where the study was conducted and where the data were analyzed. Therefore, we have added a following paragraph in the 'Methods' section.

“Clinical data were collected at each participating institution and sent to Yokohama City University for comprehensive analysis. Similarly, tests for driver oncogene mutations and PD-L1 expression were conducted at each institution using standardized and validated methods. These data were also collated at Yokohama City University for subsequent analysis.”

Comments 2: Author should add a demographic data in the form of table or chart, to assess the type of mutation associated with the area where they enrolled.

Response 2: We appreciate your suggestion to include demographic data for a more comprehensive understanding of the study. However, it's noteworthy that all participating institutions are located within the same prefecture in Japan, and all patients enrolled in this study are Japanese. Given this geographic and demographic consistency, we did not include additional demographic tables. We have, however, added this information in the 'Methods' section to clarify the context of our study, which we believe addresses the core of your recommendation.

” This is a multicenter, retrospective observational study conducted in collaboration with eight medical facilities located in Kanagawa Prefecture, Japan. It is important to note that all eight participating institutions are located within the same prefecture. The pri-mary objective of this research is to assess the efficacy and safety of various systemic therapy as a first line in patients with NSCLC. Ethical approval for the study was granted by the Ethics Committee of Yokohama City University (Approval Number: B191200044). Informed consent was waived due to the retrospective nature of the study.”

Comments 3: In result, author should add legends with figure 4A to 4D.

Response 3: I agree with you. We have included detailed legends for Figures 4A to 5D, as you advised, to provide clearer context for each subfigure.

“Figure 4A. Kaplan-Meier Curves for Overall Survival After Initiation of Molecular Targeting Agents Among Patients With Indicated Driver Oncogene Mutations. The Kaplan-Meier survival curves showing OS outcomes for patients treated with molecular targeting agents, categorized based on the presence of specific driver oncogene mutations. The subgroups include ALK, BRAF, EGFR, MET, and ROS1. Notably, no statistical difference was observed in OS among the various subgroups, despite the limited number of patients in some categories.”

“Figure 4B. Kaplan-Meier Curves for Overall Survival After Initiation of Cytotoxic Agents With or Without Immune Checkpoint Inhibitors Among Patients Without Driver Oncogene Mutations or Not Evaluated for Mutations. The Kaplan-Meier survival curves for two distinct patient groups: those who were not evaluated for driver oncogene mutations ("NE") and those without any de-tected mutations ("No mutation"). Median survival times and 95% confidence intervals are given for each group. The figure highlights the significantly superior OS outcomes for patients who were evaluated and found to have no driver oncogene mutations, compared to those not evaluated (p = 0.021).”

“Figure 5: Kaplan-Meier Curves for Overall Survival After Initiation of Chemotherapy Among Patients with Indicated PD-L1 Expression Levels

  1. Among patients treated with any form of treatment.
  2. Among patients treated with combination chemotherapy and immune checkpoint inhibitors or immune checkpoint inhibitors alone.
  3. Among patients treated with molecular targeting agents.
  4. Among patients treated with chemotherapy using cytotoxic agents.”

Comments 4: Author may elaborate discussion with observed result and existing or used other techniques used to evaluate the efficacy of therapy.

Response 4: Thank you for your insightful suggestion. In our study, we mainly employed Progression-Free Survival (PFS) and Overall Survival (OS) to evaluate the efficacy of therapy. We acknowledge that other parameters such as Objective Response Rate (ORR) and Disease Control Rate (DCR) could provide additional insights into treatment outcomes. Furthermore, Quality of Life (QOL) related to therapy is another significant metric that we were unable to assess in this study. These limitations have been added to the revised manuscript to provide a more balanced view of our research.

In line 443 – 447; “the study primarily utilized PFS and OS as the metrics for assessing therapeutic efficacy. Although these are robust endpoints, other potentially useful metrics such as Objective Response Rate and Disease Control Rate were not included. Moreover, the study did not assess therapy-related Quality of Life, another critical indicator for evaluating the effectiveness of chemotherapy regimens.”

Reviewer 3 Report

Comments and Suggestions for Authors

This manuscript is an original paper aiming to describe the patients’ outcomes to complement the growing real-world literature on the treatment of NSCLC patients. Findings highlight the importance of identifying driver oncogene mutations and administering appropriate systemic treatment to improve patients’ outcomes.

This paper is an interesting study. I appreciated the analysis conducted by the authors, but some points, mainly in the section on Material and Methods and Results, should be improved. Below, I leave some minor and major comments so that the author could consider, if they wish, a revised version of their manuscript.

1)      Please beware of abbreviations. They should be defined the first time they appear in the abstract, the main text, the first figure or the table. Write what the abbreviations ALK, ROS1, BRAF, MET, RET, PD-1 and CTLA4 stand for

2)      I suggest adding “real-word evidence” in the keywords.

3)      There is an improper use of the term chemotherapy. Chemotherapy is a type of cancer treatment different from molecular targeting or immune checkpoint inhibitors. Authors cannot use the term chemotherapy when they refer to molecular targeting agents or immune checkpoint inhibitors. Molecular targeting agents and immune checkpoint inhibitors are not chemotherapy regimens. The term chemotherapy is indicated only to describe the group C (chemotherapy with cytotoxic agent). Please replace the term chemotherapy regimens with systemic anticancer treatments or cancer regimens when referring to various types of regimens. This misinterpretation recurs throughout the text.

4)      Please report in the Material and Methods the sources (electronic/paper health records, administrative databases, structured or unstructured data (as clinical notes) from which data were extracted.

5)      Please report if a manual review supplemented data collection

6)      The inclusion criteria reported in the sentences from lines 99 to 102 are unclear, maybe for the improper use of the term chemotherapy. Please re-write the sentences better. I understand that the study population include all adult patients (aged ≥18 years) with a confirmed diagnosis of NSCLC who initiated a first-line treatment between January 2015 and December 2022

7)      Authors reported at line 103 that patients who underwent chemoradiotherapy and those receiving adjuvant chemotherapy following surgical intervention were excluded. It is unclear if these patients were always excluded or if they progressed to the first-line and subsequently received a palliative first-line were included or not. Please specify this issue in the text

8)      Please report in the text the index date (e.g. the initiation of the first-line) and the minimum potential follow-up? Until when were patients followed-up?

9)      Please use bullet points from line 107 to line 122. When the authors define the order of the systemic groups as Chemotherapy, TKI, ICIs and Chemio + Immuno for the first time in the section Materials and Methods, it is advisable to follow the same order when results are shown in tables and figures)  In Europe, Nivolumab received market access approval at the national level for the second-line, not the first-line. Also, durvalumab as monotherapy is indicated for the treatment of locally advanced, unresectable non-small cell lung cancer (NSCLC) whose disease has not progressed following platinum-based chemoradiation therapy (as maintenance therapy after a curative treatment). If in Japan, autorizations were the same, patients treated with nivolumab and durvalumab as monotherapy cannot be considered as first-line.

11)   The Statistical analysis section is incomplete. Please describe better assessments and study endpoints. How was overall survival defined (as the time period from the index date to death due to NSCL cancer or any cause)? For patients whose vital status could not be verified or without event, how the survival time was censored (end of follow-up, last contact and so on)?

12)  Please specify if the clinical stage reported in Table 1 is the tumour stage at the earliest NSCLC diagnosis

13)  Table 1 Please indicate when metastases are present (at LAM diagnosis, at inclusion date, at the beginning of the first line)

14)  Significant clinical characteristics such as smoking history or performance status (PS) are missing. Can authors retrieve these data?

15)  Please add the overall patients by treatment groups in Table1.

16)  Why did the authors estimate only the overall survival and not the progression-free survival associated with the first-line?

17)  Table 2 could be more significant if compared with the overall test conducted for each diagnostic method. Please add in Table 2 the total number of tests and the relative percentage.

18) Figure 4B Please beware of the patients’ numbers 84 and 396 used to estimate OS for the group NONE (without any detected driver oncogene mutations) and NE (not evaluated), respectively, after the initiation of chemotherapy and chemotherapy + immuno. These numbers are different if compared with the data reported in Table 1.

19)  Why multivariate analysis of factors affecting OS did not include the tumour stage or histology, which are known as potential confounders that could have influenced the present results

20)  I suggest to include the following references useful for discussion:

·         “Real-World Outcomes and Treatments Patterns Prior and after the Introduction of First-Line Immunotherapy for the Treatment of Metastatic Non-Small Cell Lung Cancer” by  Danesi V. and et al.

·         “Molecular testing and treatment patterns for patients with advanced non-small cell lung cancer: PIvOTAL observational study. by Lee, D.H.;  PLoS ONE 2018; 

Author Response

Response to Reviewer 3 Comments

1. Summary

Thank you very much for taking the time to review this manuscript. Please find the detailed responses below and the corresponding revisions in track changes in the re-submitted files.

  1. Point-by-point response to Comments and Suggestions for Authors

Comment 1: Please beware of abbreviations. They should be defined the first time they appear in the abstract, the main text, the first figure or the table. Write what the abbreviations ALK, ROS1, BRAF, MET, RET, PD-1 and CTLA4 stand for

Response: Thank you for bringing to our attention the importance of defining abbreviations upon their first appearance in the manuscript. We have revised the text accordingly to clarify all abbreviations in the abstract, main text, figures, and tables.

In line 75 – 82; “Anaplastic lymphoma kinase (ALK) [9], ROS proto-oncogene 1 (ROS1), V-raf murine sarcoma viral oncogene homolog B1 (BRAF) [10], MET proto-oncogene, receptor tyrosine kinase (MET) [11], and RET proto-oncogene receptor tyrosine kinase (RET) [12] has be-come increasingly critical in routine clinical practice.

Another pivotal development is the emergence of cancer immunotherapy. The discovery of immune checkpoint molecules such as Programmed cell death protein 1 (PD-1) and Cytotoxic T-lymphocyte-associated protein 4 (CTLA4) towards”

Comment 2: I suggest adding “real-word evidence” in the keywords.

Response: We appreciate your suggestion to include "real-world evidence" in the list of keywords. This has been duly added.

In the keywords section; “Keywords: Non-Small Cell Lung Cancer; Molecular Targeting Agents; Immune Checkpoint In-hibitors; Overall Survival; Driver oncogene; PD-L1 Expression; Personalized Medicine; Re-al-world Evidence”

Comment 3: There is an improper use of the term chemotherapy. Chemotherapy is a type of cancer treatment different from molecular targeting or immune checkpoint inhibitors. Authors cannot use the term chemotherapy when they refer to molecular targeting agents or immune checkpoint inhibitors. Molecular targeting agents and immune checkpoint inhibitors are not chemotherapy regimens. The term chemotherapy is indicated only to describe the group C (chemotherapy with cytotoxic agent). Please replace the term chemotherapy regimens with systemic anticancer treatments or cancer regimens when referring to various types of regimens. This misinterpretation recurs throughout the text.

Response: Your point regarding the correct terminology for different types of cancer treatments is well-taken. We have replaced the term "chemotherapy" with "systemic anticancer treatments" throughout the text as appropriate.

This is an example of the revised section: “Overall Survival Among Patients Treated With Different Systemic anticancer treatments”

Comment 4: Please report in the Material and Methods the sources (electronic/paper health records, administrative databases, structured or unstructured data (as clinical notes) from which data were extracted.

Response: We appreciate your suggestion to specify the sources from which data were extracted. We have revised the "Material and Methods" section to address this point. Specifically, we have added the following sentences:

"Clinical data were extracted from electronic health records at each participating institution and sent to Yokohama City University for comprehensive analysis. Similarly, tests for driver oncogene mutations and PD-L1 expression were conducted at each institution using standardized and validated methods. These data were also collated at Yokohama City University for subsequent analysis."

Comment 5: Please report if a manual review supplemented data collection

Response: We appreciate your inquiry about whether a manual review was conducted to supplement data collection. We would like to clarify that no manual review was performed for this study. All data were systematically extracted from electronic health records at each participating institution and sent to Yokohama City University for comprehensive analysis. We have added this information to the "Material and Methods" section to provide full transparency about our data collection process.

Comment 6: The inclusion criteria reported in the sentences from lines 99 to 102 are unclear, maybe for the improper use of the term chemotherapy. Please re-write the sentences better. I understand that the study population include all adult patients (aged ≥18 years) with a confirmed diagnosis of NSCLC who initiated a first-line treatment between January 2015 and December 2022

Response: Thank you for pointing this out. To address your feedback, we have modified the "Study Population" section. The revised section now clarifies that eligible individuals were those pathologically diagnosed with non-small cell lung cancer (NSCLC) and treated within the interval from January 1, 2015, to December 31, 2022. Furthermore, the inclusion criteria have been refined to specify that patients who were administered any anti-cancer medication were eligible, thereby expanding the scope beyond chemotherapy alone. We trust that these modifications sufficiently address your concerns.

In line 106 - 111 Individuals eligible for this study were those who received a pathological diagnosis of non-small cell lung cancer (NSCLC) at any of the eight participating facilities. The diagnoses and treatments were conducted within the interval extending from January 1, 2015, to December 31, 2022. The inclusion criteria are further specified to encompass patients treated with any form of anti-cancer medication without restrictions based on performance status (PS). All enrolled patients in this study are Japanese. Exclusion criteria consisted of patients with lung cancer subtypes other than NSCLC, those undergoing chemoradiotherapy, and those receiving adjuvant chemotherapy following surgical in-tervention.”

Comment 7: Authors reported at line 103 that patients who underwent chemoradiotherapy and those receiving adjuvant chemotherapy following surgical intervention were excluded. It is unclear if these patients were always excluded or if they progressed to the first-line and subsequently received a palliative first-line were included or not. Please specify this issue in the text

Response: We acknowledge the need for clearer elaboration on this matter. The exclusion criteria have been revised to specify that patients who experienced disease recurrence after chemoradiotherapy or surgical resection were included in the study.

In the study population section: “Exclusion criteria consisted of patients with lung cancer subtypes other than NSCLC, those undergoing chemoradiotherapy, and those receiving adjuvant chemotherapy following surgical intervention. However, patients who experienced disease recurrence following chemoradiotherapy or surgical resection were eligible for inclusion.”

Comment 8: Please report in the text the index date (e.g. the initiation of the first-line) and the minimum potential follow-up? Until when were patients followed-up?

Response: We appreciate the reviewer's suggestion to clarify the duration of patient follow-up in our study. To address this, we have added the following sentence in the Methods section: "The study observed patients from the initiation of their first-line treatment until the end of December 2022, serving as the final follow-up date."

Comment 9: Please use bullet points from line 107 to line 122. When the authors define the order of the systemic groups as Chemotherapy, TKI, ICIs and Chemio + Immuno for the first time in the section Materials and Methods, it is advisable to follow the same order when results are shown in tables and figures)  In Europe, Nivolumab received market access approval at the national level for the second-line, not the first-line. Also, durvalumab as monotherapy is indicated for the treatment of locally advanced, unresectable non-small cell lung cancer (NSCLC) whose disease has not progressed following platinum-based chemoradiation therapy (as maintenance therapy after a curative treatment). If in Japan, autorizations were the same, patients treated with nivolumab and durvalumab as monotherapy cannot be considered as first-line.

Response: We appreciate the detailed feedback provided by the reviewer. Bullet points have been added for better readability, and the order of systemic groups has been standardized across the manuscript.

Comment 11: The Statistical analysis section is incomplete. Please describe better assessments and study endpoints. How was overall survival defined (as the time period from the index date to death due to NSCL cancer or any cause)? For patients whose vital status could not be verified or without event, how the survival time was censored (end of follow-up, last contact and so on)?

Response: Your suggestion to provide more detail in the Statistical Analysis section has been implemented. We have expanded upon the assessments and study endpoints.

In the material and methods section: “Statistical Analysis

Clinical data were extracted from electronic health records at each participating in-stitution and sent to Yokohama City University for comprehensive analysis. Similarly, tests for driver oncogene mutations and PD-L1 expression were conducted at each in-stitution using standardized and validated methods. These data were also collated at Yokohama City University for subsequent analysis. n the current study, Overall Survival (OS) was defined as the time interval from the initiation of treatment to the date of death from any cause. For patients whose vital status could not be confirmed, the survival time was censored at the last date of known contact or survival status update. Descriptive statistics, such as means, medians, and frequencies, were calculated. Inferential statistics employed included Kaplan-Meier survival analysis, Log-rank tests, and multivariate Cox regression models. Data were analyzed using JMP version 17.0 (SAS Institute Inc., NC, US). The level of statistical significance was set at p < 0.05. Survival curves were generated using Python (version 3.9.17), with Numpy (version 1.25.2) and Lifelines (version 0.27.7) for the statistical analysis.”

Comment 12: Please specify if the clinical stage reported in Table 1 is the tumour stage at the earliest NSCLC diagnosis

Response: Thank you for your constructive comment regarding the specification of the clinical stage reported in Table 1. To address your query, we have clarified that the staging was performed according to the 8th edition of the UICC/AJCC staging system. Additionally, we noted that for patients initially staged using the 7th edition, reclassification was undertaken in line with the 8th edition criteria. Stage IV cases were collectively categorized as 'Stage IV.'

We added following statement in table 1: ”In table 1, the clinical stage of the tumors is reported according to the 8th edition of the Union for International Cancer Control (UICC) / American Joint Committee on Cancer (AJCC) staging system. For patients initially staged using the 7th edition, reclassification was performed in accordance with the updated 8th edition criteria. Cases at any Stage IV were grouped collectively and denoted simply as Stage IV.”

Comment 13: Table 1 Please indicate when metastases are present (at LAM diagnosis, at inclusion date, at the beginning of the first line)

Response: Thank you for your comment regarding the timing of metastasis assessment, as presented in Table 1. To provide clarity, we have added a note in the table indicating that the metastases were assessed immediately prior to the initiation of treatment.

Comment 14: Significant clinical characteristics such as smoking history or performance status (PS) are missing. Can authors retrieve these data?

Response: We have added information on smoking history and performance status, as these data could be retrieved in table 1.

Comment 15: Please add the overall patients by treatment groups in Table1.

Response: We appreciate the reviewer's suggestion to provide more comprehensive demographic data. We have now included the total number of individuals in each treatment group in Table 1 for clarity.

Comment 16: Why did the authors estimate only the overall survival and not the progression-free survival associated with the first-line?

Response: Thank you for your thoughtful query regarding the inclusion of progression-free survival (PFS) as an additional endpoint. While PFS is indeed frequently employed as an alternative indicator to OS, particularly in the context of first-line treatments, our study was designed with certain considerations in mind. Given the disparate nature of treatments being compared—each with potentially highly variable PFS—we chose to concentrate on OS as our primary endpoint. This decision was also informed by the overarching objective of systemic pharmacotherapy in oncological settings, which is to prolong life. OS thus served as the most direct measure of this ultimate therapeutic goal.

Comment 17: Table 2 could be more significant if compared with the overall test conducted for each diagnostic method. Please add in Table 2 the total number of tests and the relative percentage.

Response: Thank you for your valuable suggestion to add the total number of tests and relative percentages for each diagnostic method in Table 2. We have revised the table accordingly to provide a more comprehensive overview.

Diagnostic Method

EGFR

ALK

KRAS

ROS1

MET

BRAF

None

Total

Single (S)

219 (44.0%)

21 (4.2%)

3 (0.6%)

8 (1.6%)

4 (0.8%)

1 (0.2%)

242 (48.6%)

498

Oncomine Dx (ODx)

56 (29.3%)

6 (3.1%)

7 (3.7%)

2 (1.0%)

3 (1.6%)

1 (0.5%)

116 (60.7%)

191

Liquid (L)

6 (75.0%)

0 (0.0%)

0 (0.0%)

0 (0.0%)

0 (0.0%)

0 (0.0%)

2 (25.0%)

8

Others (OTH)

29 (36.2%)

3 (3.8%)

6 (7.5%)

1 (1.2%)

4 (5.0%)

1 (1.2%)

36 (45.0%)

80

Comment 18: Figure 4B Please beware of the patients’ numbers 84 and 396 used to estimate OS for the group NONE (without any detected driver oncogene mutations) and NE (not evaluated), respectively, after the initiation of chemotherapy and chemotherapy + immuno. These numbers are different if compared with the data reported in Table 1.

Response: Thank you for bringing this discrepancy to our attention. Upon review, we realized that the patient numbers in Figure 4B did not match those in Table 1 because Figure 4B actually included all patients who received systemic anti-cancer therapy, not just those who underwent chemotherapy and chemotherapy + immuno as originally indicated. To address this issue, we have revised the caption of Figure 4B for clarity.

"Figure 4B: Kaplan-Meier Curves for Overall Survival After Initiation of Anti-Cancer Systemic Therapy Among Patients Without Driver Oncogene Mutations or Not Evaluated for Mutations. The Kaplan-Meier survival curves for two distinct patient groups: those who were not evaluated for driver oncogene mutations ('NE') and those without any detected mutations ('No mutation')."

Comment 19: Why multivariate analysis of factors affecting OS did not include the tumour stage or histology, which are known as potential confounders that could have influenced the present results

Response: Thank you for your insightful query regarding the absence of tumor stage and histology in our multivariate analysis of factors affecting overall survival (OS). Your point is well taken. The reason for this exclusion was primarily based on our pre-study univariate analyses. In those preliminary evaluations, variables like the presence of bone and liver metastases showed stronger correlations with OS compared to clinical stage. As for histology, our data indicated a strong association between adenocarcinoma and the use of targeted therapies. Given that the focus of our study was to examine the impact of targeted therapy use, we prioritized including this variable in our multivariate model.

 Additionally, to mitigate the risk of statistical errors arising from multiple testing, we had to limit the number of factors included in our multivariate model. Therefore, we decided to focus on variables that showed the most robust associations in our initial analyses. We believe that our approach was methodologically sound given the aims of our study, but we acknowledge the importance of the variables you mentioned and will consider including them in future research.

Comment 20: I suggest to include the following references useful for discussion:

  • “Real-World Outcomes and Treatments Patterns Prior and after the Introduction of First-Line Immunotherapy for the Treatment of Metastatic Non-Small Cell Lung Cancer” by Danesi V. and et al.
  • “Molecular testing and treatment patterns for patients with advanced non-small cell lung cancer: PIvOTAL observational study. by Lee, D.H.; PLoS ONE 2018;

Response: We are grateful for your recommended references and have included them in the Discussion section to enrich our manuscript. I have carefully considered the reviewer's suggestion to include additional references that could enrich the discussion section of the manuscript. Accordingly, the discussion section has been revised to integrate the studies by Danesi V. et al. and Lee, D.H., adding further context and depth to our findings. These references have been appropriately cited within the text.

In discussion section: The study provides a comprehensive evaluation of the factors influencing OS among patients diagnosed with NSCLC, incorporating a wide spectrum of variables such as types of systemic therapy, driver oncogene mutations, and PD-L1 expression levels in the re-al-world settings. The data gathered across multiple institutions lend a robust foundation to our findings, which have significant implications for both clinical practice and future research. The current data offers an invaluable contribution to emphasize the revolu-tionary advancements made in this century concerning genetic screening, molecularly targeted therapies, and immuno-modulation through PD-L1 expression. This is in line with the findings of the study by Danesi V. et al., which also emphasizes the importance of real-world evidence in understanding the impact of first-line immunotherapy in treating metastatic NSCLC [16]. These innovations have not only refined our under-standing of NSCLC, but also have opened new avenues for personalized and effective treatment strategies.

            Our results underscore the fundamental role of molecular targeted therapies in improving the 5-year OS in patients with NSCLC, corroborating the findings of Lee, D.H. in the PIvOTAL observational study, which also highlights the significance of molecular testing for optimizing treatment strategies [17]. Molecular targeting agents, specifically those targeting EGFR[6,18] , ALK [19–21], ROS1 [22], and BRAF [23], have shown su-perior outcomes in clinical trials involving patients with favorable conditions, surpassing the results achieved by standard therapies.

Round 2

Reviewer 3 Report

Comments and Suggestions for Authors

I appreciate your efforts to improve the paper.